# Study protocol for a prospective cohort study identifying risk factors for sport injury in adolescent female football players: the Karolinska football Injury Cohort (KIC)

Ulrika Tranaeus [1,2] Nathan Weiss [2,3] Victor Lyberg,[3] Martin Hagglund [4,5,6] Markus Waldén [5,6,7,8] Urban Johnson [9] Martin Asker [2,3,10] Eva Skillgate [2,3]

For numbered affiliations see end of article.

**Correspondence to**
Dr Ulrika Tranaeus;
ulrika.tranaeus@gih.se

## ABSTRACT

**Introduction** Football is a popular sport among young females worldwide, but studies concerning injuries in female players are scarce compared with male players. The aim of this study is to identify risk factors for injury in adolescent female football players.

**Methods and analysis** The Karolinska football Injury Cohort (KIC) is an ongoing longitudinal study that will include approximately 400 female football academy players 12–19 years old in Sweden. A detailed questionnaire regarding demographics, health status, lifestyle, stress, socioeconomic factors, psychosocial factors and various football-related factors are completed at baseline and after 1 year. Clinical tests measuring strength, mobility, neuromuscular control of the lower extremity, trunk and neck are carried out at baseline. Players are followed prospectively with weekly emails regarding exposure to football and other physical activity, health issues (such as stress, recovery, etc), pain, performance and injuries via the Oslo Sports Trauma Research Center Overuse Injury Questionnaire (OSTRC-O). Players who report a substantial injury in the OSTRC-O, that is, not being able to participate in football activities, or have reduced their training volume performance to a moderate or major degree, are contacted for full injury documentation. In addition to player data, academy coaches also complete a baseline questionnaire regarding coach experience and education.

**Ethics and dissemination** The study was approved by the Regional Ethical Review Authority at Karolinska Institutet, Stockholm, Sweden (2016/1251-31/4). All participating players and their legal guardians give their written informed consent. The study will be reported in accordance with the Strengthening the Reporting of Observational studies in Epidemiology. The results will be published in peer-reviewed academic journals and disseminated to the Swedish football movement through stakeholders and media.

## Strengths and limitations of this study

► A strength is the bio-psychosocial and multi-professional perspective of the risk of injuries in young female football players and factors of importance for not being injured, even though the bio-psychosocial factors are not equal included.

► Strengths are also the large sample size and the robust data collection of exposures, potential confounding factors, potential effect measure modifiers and outcomes.

► A potential limitation is the risk of misclassification of time varying exposures and outcomes in the weekly self-reported data collection.

► Using emails and SMS for weekly reports might decrease the response rates and thereby increase the risk of selection bias in the results. If the response turns out to be low, there is a risk of selection bias in the risk analyses.

## INTRODUCTION

Four million females worldwide are registered football players, of which 2.5 million are under 17 years old according to Fédération Internationale de Football Association.[1] Studies regarding injuries in female football players are fewer compared with the number of studies in male football players.[2–4] In brief, these studies show that common injuries in female football players are joint and ligament injuries to the knee and ankle joints as well as muscle and tendon injuries of the thigh. In addition, there is a particular concern for concussions and anterior cruciate ligament injuries in female players.[3 5–8]

Female football players have more absence days from football due to injuries compared with male players,[8] and long-term consequences of injuries might be considerable

for young football players.[9] For players with a history of injury, the risk of osteoarthritis in lower extremity joints are high and greater than in the general population.[10 11] Injuries may also lead to premature career ending,[12] and mental health problems.[13] Identifying risk factors for injury is, therefore, an important step towards reduction of injury risk.[14] To identify possible risk factors, well-designed prospective cohort studies are needed.[15 16] Specifically, the suggested risk factors in this setting can be classified as bio-psychosocial factors (see Wiese-Bjornstal for bio-psychosocial view on a sport injury risk profile).[17] Biological risk factors for injury in female players are previous injury,[7 18–20] a hamstring/quadriceps ratio of less than 55%, increased body mass index, as well as results of plyometric tests, for example, poor performance in drop jump landing test is associated with increased risk of ankle injury.[21] Other biological risk factors associated with an increased risk of injury during the season are young age,[6 18] physical complaints at the beginning of the season,[18] familial disposition such as a parent/sibling[18] or a twin[22] with knee injury, and lower level of preseason aerobic fitness.[23 24] Findings regarding joint hypermobility as a risk factor in female players are inconclusive in older studies,[23 25] although no association was shown in more recently published studies.[26 27] Risk factors for back pain in adolescents include rapid growth rate, and tight muscle imbalance,[28] but risk factors for football related back/neck injuries in young females are not known. Psychological risk factors reported includes somatic trait anxiety, mistrust, ineffective coping,[29] life event stress[30] and perceived mastery climate.[20] Social factors that influenced the risk for injury in female athletes are coaches' and player's education regarding injury prevention strategies,[31] stress from teammates and coaches,[20 29 32] and for back pain in adolescents; smoking.[28] In football, an identified situational specific risk factor is the playing positions defenders and strikers.[19]

In summary, most knowledge about risk factors for injuries in adolescent female football players consists of isolated factors, and lack of using multidisciplinary and bio-psychosocial perspectives. Hence, the overall aim of the Karolinska football Injury Cohort study (KIC) is to identify risk factors for injuries in adolescent female football players from a bio-psychosocial perspective. Specific aims are to determine the incidence of injuries in young female football players and to identify modifiable risk factors for such injuries. Finally, our aims include to describe changes in muscle strength and range of motion (ROM) over a year, trajectories of pain and to identify important factors for not being injured over a year.

## METHODS AND ANALYSIS

This is a prospective observational cohort study designed in agreement with Strengthening the Reporting of Observational studies in Epidemiology (STROBE) guidelines.[33]

### Study setting and participants

Football clubs with adolescent female academy players aged 12–19 years, participating in Swedish divisions 1–2 for girls in the largest regions, are eligible to participate in the study. Most players will be recruited in Stockholm. The district of Stockholm consists of 140 teams and approximate 2520 female players, 13–19 years old. Clubs which meet the inclusion criteria are contacted and invited to participate and are given oral and written information. Clubs which choose to take part in the study are provided with a more detailed oral and written information in the presence of players, legal guardians and coaches.

A cohort of approximately 400 adolescent academy players will be recruited. An internal pilot study of 63 football players has been conducted to test the infrastructure and the implementation of the study, with satisfactory results (unpublished data).

### Baseline measurements
#### Questionnaires

The baseline questionnaire covers potential risk factors for the aetiology of sport injuries as well as information about players' general health status. Players are surveyed in various areas, including *health*: health problems (eg, illness), medication, age at menarche, amenorrhoea; *lifestyle*: sleep patterns, eating habits, food supplements, tobacco as smoking or Swedish snus (snuff) and alcohol; and *socioeconomic* factors: guardians' education. Included *football-related factors* are: training and match play exposure, playing position, dominant limb, years of experience, other sports participation, injury preventive strategies (eg, the Swedish injury prevention warmup programme Knee Control)[34] and type of turf at the home facilities (artificial or natural grass) according to guidelines for football studies.[35–37] *Psychosocial factors* are surveyed using: the modified General Health Questionnaire-12 consisting of 12 items regarding self-reported general psychological health using a four-point Likert scale,[38] coping assessed by a 28 item self-report questionnaire that measure effective and ineffective strategies to cope with stressful events using a four-point Likert scale (Brief COPE),[39] player's passion to sport measured in harmonious and obsessive passions using a 14 item questionnaire with a seven-point Likert scale (Passion scale),[40] education in sport psychology, regularly seeing a sport psychologist/mental coach and perceived stress (single item question).[41] *Injury history*: injuries occurring within the previous 6 months prior to inclusion are captured using a modified Swedish version of the validated psychometric instrument Oslo Sports Trauma Research Center Overuse Injury Questionnaire (OSTRC-O).[42 43] *Back and neck pain* is covering the frequency, intensity, disability of low back pain (LBP) and upper back pain/neck pain (UBNP) and corresponding longitudinal trajectories the preceding 6 months using modified versions of the Chronic Pain Questionnaire[44] and Visual Trajectories Questionnaire-Pain,[45] respectively.

In addition, coaches in the included teams are surveyed regarding their education, years of experience, the use

of warmup and stretching regime and implementation of injury prevention programmes.

## Physical test protocol

The physical test protocol includes several tests that are considered valid, reliable, and field friendly; performed in approximately 60 min/player. The protocol comprises measurements of strength, mobility and control of lower extremity, trunk and neck and also include anthropometric measurements (height, weight and leg length). The protocol is briefly outlined below, however, further details including visual presentations is available in the electronic supplementary file (online supplemental file 1).

All test procedures are conducted in indoors facilities during weekends. The physical tests are divided into nine test stations with 1–2 test leaders each (online supplemental file 1). Hitherto, 52 clinically experienced test leaders have been involved in data collection. They were trained by MA, VL, NW and the previous test leader in charge of the station to ensure consistent execution and reliability. Information and instructions given to the players regarding the tests are standardised, and test leaders refrain from coaching or encouraging the players in any way during the procedures.

A maximum of nine players are tested per session (ie, one at each station) and are informed to train and compete as usual prior to testing. Players are informed to refrain from certain tests that evoke pain, provoke ongoing injuries or other health-related issues. Prior to performing the physical tests, players complete a standardised 7 min warm-up programme comprising 4 min of jogging, 10×1 body weight squats, 10×1 body weight squat jumps and 10×1 unilateral body weight lunges. Following the warm-up session, players are randomly assigned to a starting test station and subsequently follow a predefined order.

## Calf heel raises

Ankle plantarflexion (PF) muscle endurance is investigated using unilateral weight bearing calf heel raises.[46] The player is instructed to perform maximum unilateral barefoot heel raises continuously to failure, guided by a metronome to standardise the pace (1 s concentric, 1 s eccentric contraction). The test leader registers the number of accomplished repetitions and discontinues when the player fails to reach the marked target height. The same procedure is then conducted on the opposite foot.

## Active PF mobility

Active PF ROM is measured with a universal goniometer in supine position using fibula and fifth metatarsal as reference marks.[47 48] The player is instructed to maintain extended knees throughout the movement, and to perform a sequence of six maximal active PF cycles from a neutral dorsiflexion (DF) position, of which the final three trials are registered.

## Weight bearing ankle DF mobility

Weight bearing ankle DF ROM is measured in a lunge position with the player's foot placed on a metric ruler 10 cm away from a wall.[46 49] The player is instructed to lunge forward, until contact with the wall is achieved without allowing the heel to lift off the ground. Three warm-up trials are performed from the 10 cm mark to familiarise the player with the test. Thereafter, the test leader measures the following three trials. From the 10 cm reference mark, the player progresses 1 cm away at a time from the wall until unable to perform a successful repetition. If unable to perform a successful repetition at the 10 cm reference mark, she is asked to progress 1 cm forward until able to complete a successful repetition. The maximal DF ROM is measured with a digital inclinometer (Clinometer, Plaincode, Stephanskirchen, Germany) and distance from the wall to the greater toe is measured in centim.

## Trunk mobility

Trunk rotation mobility is measured in a modified seated rotation test, and in a lunge position on a gym mat graded with 5° increments.[50–52] The player is instructed to maximally rotate alternating between right and left, in a cross-legged position and subsequently in a lunge position on the dominant, and non-dominant limb while the test leader measures the rotational degrees in the end range. Three repetitions are performed in each direction during the three separate positions, and the mean value for each position is later used for analysis.

## Trunk strength

Isometric trunk rotational strength is measured in a modified standing wood chopper test using a force gauge to evaluate force output (RS Pro Digital Force Gauge, RS Components, Corby, UK).[53–55] In this modified test, the player holds a handle attached to the force gauge in shoulder height in a standing position. The player is instructed to generate force through her trunk and rotate for 5 s while maintaining straight arms. Three consecutive repetitions are conducted in each direction and the maximal force output is later used for analysis.

## Deep neck flexor endurance

Deep neck flexor muscle endurance is assessed through a modified version of the cranio-cervical flexion test with a pressure sensor (Stabilizer Pressure Bio-Feedback, Chattanooga Group, Hixon, Tennessee, USA).[51 56 57] The test consists of a pretest and an endurance test. In the pretest the player is positioned in a supine position on an examination table and are instructed to slightly push the neck against the pressure sensor to increase the pressure and then maintain the pressure for 3×3 s, with a 3 s rest in between each contraction, at a specific target pressure (TP), starting at 20 mm Hg. If the player can perform this task, she is instructed to increase the pressure to 24 mm Hg and keep the pressure for another 3×3 s. This is repeated with a 2 mm Hg increase until the

player reaches 30 mm Hg. If the player can perform the pretest the endurance test is subsequently performed. During the endurance test, the same setup and procedure as in the pretest is carried out. However, the player is instructed to hold each contraction at the TP for 3×10 s with a 10 s rest in between contractions. The highest completed TP with a full set of 3×10 s contractions is later used for analysis.

### Hip and knee strength

Isometric hip flexion, extension, adduction and abduction strength as well as eccentric hip abduction and adduction strength are measured with a hand-held dynamometer (HHD) (MicroFet2, Hoggan Health Industries inc. West Jordan, Utah, USA).[58 59] Furthermore, isometric knee extension strength is measured with an HHD and the player in a seated position with the knee joint in 90° of flexion. Prior to executing the strength tests, two submaximal isometric contractions in each direction are performed to familiarise the player with the procedures. Three isometric contractions with gradually increasing power output for 5 s, and three maximally eccentric contractions for 3 s are performed in the isometric and eccentric tests, respectively, with a 10 s rest in between contractions. The maximal power output for each position is later used for analysis.

### Hip mobility

Measures of passive hip ROM in flexion and abduction in prone position and extension, internal and external rotation in supine position is obtained using a universal goniometer.[60 61] Three consecutive measurements for each position are performed for both the dominant and the non-dominant leg, and the mean value for each position is later used for analysis.

### Functional performance tests

To assess the player's unilateral jump performance, the One-leg Long Box Jump Test (OLLBJ) and square hop test are performed.[62 63] A 40×40 cm square is marked on the foundation and later used as a reference mark in both tests.

In the OLLBJ, the starting position are calculated by dividing the player's height (cm) with 1.6 (height/1.6). Thereafter, the player is instructed to stand on one leg on the starting position and then jump on one leg directed inside the boundaries of the square and maintain balance after landing. Three warm-up trials and five consecutive test trials are performed on each leg. The total number of approved trials are registered by the test leader.

During the square hop test, previously described in detail,[62 63] the player is instructed to jump on one leg in and out of the square as many times as possible for 15 s in a clockwise direction, timed with a stopwatch while the test leader registers the number of approved jumps. The player performs two warm up trials on each foot prior to executing the test.

### Ankle and knee stability

To assess stability of player's talocrural joints, a modified anterior drawer test is employed.[64 65] Furthermore, a modified version of Fairbank's apprehension test is used to evaluate the player's stability in the patellofemoral joint.[66] The tests are conducted on both the dominant and non-dominant foot and knee and are considered positive if the player experience any pain or discomfort during the examination, and/or an involuntary contraction of the quadriceps musculature occur during the Fairbank's apprehension test.

### Isometric back extensor endurance

Isometric back extensor endurance is assessed by the modified Sorensen test.[67–69] In this previously described modified test,[67 68] the player's lower body is supported by an examination table in prone position with three straps and the anterior–superior iliac spine is aligned with the edge of the table. The player is instructed to keep her arms folded across the chest throughout the procedure and isometric maintaining the upper body in a horizontal position until failure. The test leader registers the time elapsed until failure. A digital inclinometer (Clinometer, Plaincode, Stephanskirchen, Germany) is placed on a metric ruler at the level of the 5th vertebra in the thoracic spine to monitor sagittal plane movement. Prior to the assessment, the player completes a shorter warmup trial to orient the desired sagittal plane target angle.

### Follow-up measurement and outcome

Follow-up measurements are collected prospectively during 1 year from the baseline. In the weekly online questionnaire, the players are asked to answer questions regarding new and ongoing injuries, LBP and UBNP intensity, social support, perceived stress, recovery, and to be able to consider workload, number of training and match play hours/week.[70] To assess whether players sustain football related injuries throughout the follow-up period, the Swedish version of OSTRC-O is employed and included in the weekly online questionnaire.[42 43 71] Two study specific adaptions were made to the OSTRC-O. First, a question regarding absence/reduced participation in training/match due to reasons not related to injuries was added. Second, the option to specify injuries in different anatomical localisations in the lower and upper extremity, back, neck, head and abdomen was included.

Football related injuries reported with the OSTRC-O in the weekly online questionnaire leading to moderate or severe reductions in participationand/ or sports performance or complete inability to participate in sport are classified as a substantial injury in this study.[42] Players reporting new substantial injuries are contacted via telephone by a clinically experienced research assistant to answer a standardised interview with questions concerning the injury such as: injury mechanism, localisation, type, time-loss, re-injury, diagnosis and medical care. Injuries are divided into acute and gradual onset. An acute injury is defined as a result from a specific,

identifiable event, whereas injuries with gradual onset are defined as an injury without a single, identifiable event responsible for the injury.[35] Players receive an automated link to the online questionnaire sent by email each Sunday, with a reminder email the next day to players not responding. Furthermore, if no response is received, a text message reminder with the link is sent on Tuesdays. Finally, every other week representatives of the study visit participating football clubs to collect unanswered surveys for the previous 2-week period.

After 52 weeks of participation, a questionnaire with equivalent content as the baseline questionnaire (excluding OSTRC-O with 2 months and 3–6 months recall) are distributed to the players to evaluate possible changes from the baseline characteristics. The first 106 included players also underwent a secondary physical test protocol after 52 weeks of follow-up. In the 1-year follow-up questionnaire, different aspects of UBP and LBP, respectively, in the preceding 6 months are measured. 'Have you had UBP/Have you had LBP' (Yes/No)? If yes, has the pain hindered your daily activities (No, Yes to some extent or Yes to a high degree)? If Yes, the 'VTQ-P' is used to capture the longitudinal state of a player's pain experience of UBP and LBP and are retrospectively reported for the preceding 6-month period.[45] See table 1 for an overview of the measurements during the different phases of the study.

### Sample size

The statistical power for the analyses will depend on the exact research question, the number of exposed players and whether the exposure is continuous or categorised. The sample size in the KIC project is based on the definition 'a substantial injury' as proposed by Clarsen et al[42] and back injuries in adolescent female players in a previously published study.[7] Based on a relative risk of 1.9 for a substantial injury in the back/neck, when 88 of the players are exposed, and with a power of 0.80, a significance level 5% and with potential 10% drop out and a follow-up time of 1 year to identify risk factors, 420 players will be included.

| Table 1 | Summary of the included measurements during the different phases of the study | |
|---|---|---|
| **Phase** | **Measurements** | **Tests/tools** |
| Baseline: players (consecutive during inclusion; 2016–ongoing) | Demographic information, general health status (history of pain, illness, medication, plagues, menstrual cycle, back and neck pain), lifestyle (sleep patterns, resilience, food supplements, use of tobacco or alcohol), stress, socioeconomic factors (guardians' education), football related factors (position, years of experience, injury preventive strategies) | KIC Baseline players, The Chronic Pain Questionnaire (CPQ),[44] Visual Trajectories Questionnaire-Pain (VTQ-P)[45] |
| | Anthropometric measurements (height, weight, leg length), and measurement of strength, mobility and control of lower extremity, trunk, and neck | KIC test protocol |
| | History of injury and complaints | Modified OSTRC-O[42 43] |
| | Passion | Passion scale[40] |
| | General health | GHQ-12[38] |
| | Coping strategies | Brief COPE[39] |
| Baseline: coaches (consecutive during inclusion; 2016–ongoing) | Education, years of experience, the use of warmup and stretching regime and implementation of injury prevention programmes | KIC Baseline coaches |
| Weekly follow-up: players (September 2016–ongoing) | Exposure to football training and match play | KIC weekly report |
| | Exposure to other physical activity | |
| | Health (eg, stress, recovery) and social support | |
| | Report on pain, injury performance, complaints | Modified OSTRC-O[42 43] |
| In case of a substantial injury event | Report on injury/complaint (type of injury, localisation, inciting event) | KIC medical report |
| One-year follow-up: players (consecutive after 52 weeks participation: 2017–ongoing) | Football related factors (position, injury preventive strategies). Health status (pain in back or neck) lifestyle (sleep patterns, resilience, food supplements, use of tobacco or alcohol, physical activity), stress, coping and passion for sport | KIC 1-year questionnaire |
| One-year follow-up (consecutive after 52 weeks participation in the first 106 included players) | Anthropometric measurements (height, weight, leg length), and measurement of strength, mobility and control of lower extremity, trunk and neck | KIC test protocol |

GHQ-12, General Health Questionnaire-12; KIC, Karolinska football Injury Cohort; OSTRC-O, Oslo Sports Trauma Research Center Overuse Injury Questionnaire.

## Statistical methods

The data in the KIC study will be used to answer several different research questions and therefore, different analyses methods and statistics will be used. Kaplan-Meier estimates will be used to describe incidence. Cox regression analyses or discrete time survival analyses will be used to measure the associations between exposure and outcome, and to adjust for confounding. Only players without substantial injuries the two preceding months (reported in the baseline questionnaire) will be considered in the risk analyses, and stratified analyses to examine effect measure modification will be performed when relevant. The development of injuries is likely complex. This justifies why we measure an extensive number of factors so that we can consider confounders, intermediators and effect measure modifier in these analyses. When identifying trajectories of time, and various factors generalised estimating equations will be used for these analyses to consider the covariance between repeated measurements.

## Time plan

Players will be recruited from 2016 and followed weekly for 1 year regarding injuries/complaints. Players will consecutively be invited and included from the year they turn 13 years old and play in a participating club. The inclusion of participants will continue until we reach over 400 players.

## Data statement

The dataset and statistical codes will be available on reasonable request when the data collection is completed.

## Patient and public involvement

No patient involved.

## ETHICS AND DISSEMINATION

The study was approved by the Regional Ethical Review Authority at Karolinska Institutet, Stockholm, Sweden (2016/1251-31/4). All participating players and their legal guardians receive written and oral information regarding the study and give their written informed consent when entering the study. Players under the age of 15 are required to have written informed consent from their legal guardians. The study will be performed in accordance with the recommendations guiding research involving human subjects adopted by the 18th World Medical Association General Assembly, Helsinki, Finland, June 1964, amended at the 64th World Medical Association General Assembly, Fortaleza, Brazil, October 2013. The study will be reported in accordance with the STROBE.[33] The results will be presented in scientific conferences and published in peer-reviewed academic journals as well as being disseminated to the Swedish football movement through stakeholders and media.

**Author affiliations**
[1]Swedish School of Sport and Health Sciences, Stockholm, Sweden
[2]Institute of Environmental Medicine, Karolinska Institutet, Stockholm, Sweden
[3]Department of Health Promotion Science, Sophiahemmet University, Stockholm, Sweden
[4]Department of Health, Medicine and Caring Sciences, Linköping University, Linkoping, Sweden
[5]Sport Without Injury ProgrammE (SWIPE), Linköping University, Linkoping, Sweden
[6]Football Research Group, Linköping, Sweden
[7]Department of Orthopaedics, Hässleholm-Kristianstad Hospitals, Hässleholm, Sweden
[8]Unit of Community Medicine, Department of Health, Medicine and Caring Sciences, Linköping University, Linköping, Sweden
[9]Halmstad University, Halmstad, Sweden
[10]Scandinavian College of Naprapathic Manual Medicine, Stockholm, Sweden

**Acknowledgements** Martin Samuelsson, Lena Holm and Henrik Källberg have been involved in planning of study design, Joakim Bogren, Victor Ramirez Kristiansen, contributed to the first version of the draft. Students from the classes DN 47, 48, 49, 50, 51 and 53 at the Scandinavian college of Naprapathic Manual Medicine, Stockholm, Sweden are participating during inclusion of participants, the physical tests and the prospective data collection.

**Contributors** VL, ES, MA and UT initiated the study. All authors, UT, NW, VL, MH, MW, UJ and ES conceived the study and contributed to the development of the study protocol. ES is the study guarantor. UT and NW bi-drafted the manuscript which was critically revised in steps by all coauthors. The final manuscript was approved by all authors.

**Funding** This study is funded by grants from Swedish Research Council for Sport Science (grant number: P2019-0045, P2020-0100). The Swedish Naprapathic, and the Norwegian Naprapathic Associations, Active life foundation and Sophiahemmet foundation. There are no grant numbers for the latter funders (award/grant number: N/A).

**Competing interests** None declared.

**Patient consent for publication** Not applicable.

**Provenance and peer review** Not commissioned; externally peer reviewed.

**ORCID iDs**
Ulrika Tranaeus http://orcid.org/0000-0002-2102-6352
Nathan Weiss http://orcid.org/0000-0002-1814-020X
Martin Hagglund http://orcid.org/0000-0002-6883-1471
Markus Waldén http://orcid.org/0000-0002-6790-4042
Urban Johnson http://orcid.org/0000-0003-0990-4842
Martin Asker http://orcid.org/0000-0002-5387-3572
Eva Skillgate http://orcid.org/0000-0003-2096-1530

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
