## [Reviewer comments · BMJ Open]

ARTICLE DETAILS

TITLE (PROVISIONAL)	Study protocol for a prospective cohort study identifying risk factors for sport injury in adolescent female football players: the Karolinska football Injury Cohort (KIC)
AUTHORS	Tranaeus, Ulrika; Weiss, Nathan; Lyberg, Victor; Hagglund, Martin; Waldén, Markus; Johnson, Urban; Asker, Martin; Skillgate, Eva

VERSION 1 – REVIEW

REVIEWER	Malcolm, Dominic Hospital for Special Surgery, Academic Training
REVIEW RETURNED	21-Aug-2021

GENERAL COMMENTS	At a couple points in the paper, you note that there have not been many studies on female footballers. I think there have been plenty of studies that show that female football players are at increased risk for ACL injuries. So I don't know if it's fair to say that studies on females are scarce. I recommend that you try to capture prior injuries (specifically knee and lower extremity) even more than 2 months before enrollment. You should also consider recording time of first menses (to better understand how skeletally mature each subject is during the study). As part of your enrollment examination, you should also try to capture lower extremity alignment parameters (e.g. Q-angle) and laxity (e.g. Beighton score) When will recruitment start? This says 2017 at one point, but I assume that just hasn't been updated. In terms of communications and recording follow-up surveys, weekly communications might be onerous. Biweekly or monthly communications might be better than weekly. I think SMS messages are a good way to engage the younger patients. Also, consider engaging the parents in these messages as well (a response from a parent or a subject can give you another good data point). The paper also should be reviewed for clarity in the English language. There are several sentences that run-on or otherwise need revision. These instances are numerous, so I respectfully suggest having a native English writer help with the revision. I think a revision for English language clarity would benefit this paper. Otherwise, it seems like the authors are planning to have a very comprehensive evaluation of the players in their cohort and
--

	follow them longitudinally to capture a variety of injuries. I hope that their adolescents reliably fill out the forms at regular intervals (as I don't know if American children would reliably fill out weekly surveys!).
--	---

REVIEWER	Brewer, Britton Springfield College
-----------------	--

REVIEW RETURNED	30-Aug-2021
-------------

GENERAL COMMENTS	In this manuscript, a protocol for a prospective study of risk factors for injury among adolescent female football players is presented. The manuscript is topically appropriate for this journal and an attempt to examine antecedents of sport injury in an underinvestigated population is a worthy endeavor. These positive impressions notwithstanding, I have several concerns about the manuscript in its current form:  1. Although it is implied that a biopsychosocial perspective has been adopted in describing the risk factors to be examined in the proposed investigation, a theoretical framework for organizing relations among the biopsychosocial factors is not articulated. 2. In the summary beginning on the second-to-the-last line of page 5, there is a reference to "inconsistent knowledge." Because few inconsistencies are identified in the preceding paragraphs, it is unclear exactly what constitutes the "inconsistent knowledge." 3. The statement on the last line of page 5 regarding the "lack of using a bio-psychosocial perspective in research" is contradicted by abundance of psychological and social factors identified in the preceding paragraph. 4. How many clubs and players are in the first and second Swedish divisions? What percentage of the population will the 400 participants constitute? 5. Will any incentives be provided to clubs and/or players to participate in the study? Given the longitudinal design and respondent burden, incentives could be useful for increasing the quantity and quality of the data obtained. 6. Inclusion of the detailed descriptions (and photographs) of the physical tests is very helpful. It would also be helpful to include the questionnaires used to assess the other (e.g., injury history, psychological, social) constructs, as they are not described in detail. 7. How susceptible are the physical tests to training-related change? Do test values typically change over the course of the season? If so, they should be measured multiple times to obtain greater sensitivity with respect to vulnerability to injury. 8. Is the single-item perceived stress scale sufficient to account for the "history of stressors" variable that has been so consistently related to the occurrence of sport injury in previous research? 9. How does the multitude of variables assessed and the prospect of many interactions affect statistical power and sample size calculations?
--

	10. With reference to point 1 above, to what extent will a theoretical framework be used to guide and organize the statistical analysis? Will the statistical analysis take into account the “complex systems approach” of Bittencourt et al. (2016)? Bittencourt, N. F., Meeuwisse, W. H., Mendonca, L. D., Nettel-Aguirre, A., Ocarino, J. M., & Fonseca, S. T. (2016). Complex systems approach for sports injuries: Moving from risk factor identification to injury pattern recognition – narrative review and new concept. British Journal of Sports Medicine, 50, 1309-1314. doi:10.1136/bjsports-2015-095850
--	---

REVIEWER	Malcolm, Dominic University of Loughborough
REVIEW RETURNED	20-Sep-2021

GENERAL COMMENTS	Overall this is a well presented study protocol which clearly describes the work being undertaken. The work requires a more careful proofread as there are a number of sentences which lack clarity. There is no discussion of how the study is funded. There is no discussion of the limitations of this study Finally - and in part these comments relate to what I see as a limitation of the study, but make a more substantive, standalone point - I feel that the description of the study as ‘bio-psychosocial’ is a misrepresentation. For instance, within the list of ‘social factors that influenced the risk of injuries in female athletes’ (page 5) are 3 references to psychological studies of stress (references 20, 29 and 32). The other factors are player/coach education, smoking and playing position. Subsequently, in the description of the questionnaires used in the KIC there is only one factor (guardians’ education) listed under socioeconomic factors while ‘tobacco as smoking’ is listed under lifestyle and playing position is listed under ‘football-related factors’. My point is that the ‘social’ in ‘bio-psychosocial’ is not treated with the same level of rigor and depth as the ‘bio’ or the ‘psycho’. You could go into detail about the social determinants of, eg. diet and tobacco smoking (both strongly influenced by ethnicity, gender and social class), but that does not seem to be part of the study. For me the lack of a thorough treatment of the social is a limitation of the study. The nature of reviewing protocols is such that the only recommendation can be to re-describe the project. Ensure that what you describe as social really is social, and ensure consistency in describing the social. It should be fundamental to any approach describing itself as bio-psychosocial.
---

VERSION 1 – AUTHOR RESPONSE

Reviewer: 1

Dr. Christopher DeFrancesco , Hospital for Special Surgery, Comments to the Author:

At a couple points in the paper, you note that there have not been many studies on female footballers. I think there have been plenty of studies that show that female football players are at increased risk for ACL injuries. So, I don't know if it's fair to say that studies on females are scarce.

Response: Thank you for commenting this. We are aware of the studies regarding ACL-injuries in female football players. We have also included studies in female football players in the Introduction. Besides these articles, there is considerably fewer studies in female players than in male players and even fewer in young female players as also highlighted in recently published reviews (see below). We have also noticed a lack of prospective longitudinal studies capturing all kinds of injuries. This study is aiming to fill this gap.

Robles-Palazón FJ, López-Valenciano A, De Ste Croix M, Oliver JL, Garcia-Gómez A, Sainz de Baranda P, Ayala F: Epidemiology of injuries in male and female youth football players: A systematic review and meta-analysis. *J Sport Health Sci* 2021:in press

Okholm Kryger K, Wang A, Mehta R, Impellizzeri F, Massey A, McCall A: Research on women's football: a scoping review. *Sci Med Football* 2021:in press

I recommend that you try to capture prior injuries (specifically knee and lower extremity) even more than 2 months before enrollment.

Response: We agree that two months is short, and our baseline questionnaire therefore already cover injuries the past 6 months divided into the last two months and the last three-six months. Please, see page 7 for information. We would argue that self-reporting of prior injury history from farther back than 6 months involves a high risk of recall bias.

You should also consider recording time of first menses (to better understand how skeletally mature each subject is during the study).

Response: We agree this is important and already have a question in the baseline questionnaire regarding age at menarche. Please, see page 7 for information.

As part of your enrollment examination, you should also try to capture lower extremity alignment parameters (e.g. Q-angle) and laxity (e.g. Beighton score)

Response: Thank you for these suggestions, we agree that these factors are important. However, our aim is to focus on injuries in all body parts, and studies on associations with, for example, laxity and malalignment have been conducted previously. We therefore unanimously decided not to include these tests in this study and rather put our focus on less studied variables.

When will recruitment start? This says 2017 at one point, but I assume that just hasn't been updated.

Response: The data collection started in a smaller scale already in 2017 so this is correct, but the submission of this protocol has been delayed mainly due to lack of funding and the pandemic. Importantly, however, there are no changes in the study design or in the data collection procedures since inception.

In terms of communications and recording follow-up surveys, weekly communications might be onerous. Biweekly or monthly communications might be better than weekly. I think SMS messages are a good way to engage the younger patients. Also, consider engaging the parents in these messages as well (a response from a parent or a subject can give you another good data point).

Response: Thank you for your suggestions. We have carefully considered different ways to engage young girls such as using e-mail, SMS or a smartphone app, etc. Our previous experience on, for example, young handball players in Sweden (Asker et al., BMC Musculoskeletal Disord, 2017) is that weekly e-mails with reminding SMS gave high response rates so we decided to stick to the same approach here. We will keep your suggestions regarding involving the parents in mind for future projects.

The paper also should be reviewed for clarity in the English language. There are several sentences that run-on or otherwise need revision. These instances are numerous, so I respectfully suggest having a native English writer help with the revision.

Response: Thank you for highlighting this. We have revised the language and some of the changes are marked with “track changes”.

I think a revision for English language clarity would benefit this paper. Otherwise, it seems like the authors are planning to have a very comprehensive evaluation of the players in their cohort and follow them longitudinally to capture a variety of injuries. I hope that their adolescents reliably fill out the forms at regular intervals (as I don't know if American children would reliably fill out weekly surveys!).

Response: Thank you. We have had the revised manuscript reviewed by an English native researcher.

Reviewer: 2

Dr. Britton Brewer, Springfield College, Comments to the Author:

In this manuscript, a protocol for a prospective study of risk factors for injury among adolescent female football players is presented. The manuscript is topically appropriate for this journal and an attempt to examine antecedents of sport injury in an underinvestigated population is a worthy endeavor. These positive impressions notwithstanding, I have several concerns about the manuscript in its current form:

1. Although it is implied that a biopsychosocial perspective has been adopted in describing the risk factors to be examined in the proposed investigation, a theoretical framework for organizing relations among the biopsychosocial factors is not articulated.

Response: The biopsychosocial perspective adapted to the identification of risk factors for sport injury in adolescent female football players works as an underlying perspective for the directions of the research design. More specifically, this perspective guided the selection of potential risk factors from the three different disciplines (biological, psychological, and social). As for now, we have no hypothesis for the interactions regarding the different risk factors. Our aim is to analyse our data to be able to describe potential risk factors using, in a holistic and multidisciplinary perspective, e.g., confounders and directed acyclic graphs, DAG.

2. In the summary beginning on the second-to-the-last line of page 5, there is a reference to “inconsistent knowledge.” Because few inconsistencies are identified in the preceding paragraphs, it is unclear exactly what constitutes the “inconsistent knowledge.”

Response: Thank you for commenting this mistake. The sentence is re-written:

In summary, most knowledge about risk factors for injuries in adolescent female football players consists of isolated factors, and lack of using multidisciplinary and bio-psycho-social perspectives

3. The statement on the last line of page 5 regarding the “lack of using a bio-psycho-social perspective in research” is contradicted by abundance of psychological and social factors identified in the preceding paragraph.

Response: The sentence is re-written.

In summary, most knowledge about risk factors for injuries in adolescent female football players consists of isolated factors, and lack of using multidisciplinary and bio-psycho-social perspectives

4. How many clubs and players are in the first and second Swedish divisions? What percentage of the population will the 400 participants constitute?

Response: Most players will be recruited in Stockholm. The district of Stockholm consists of 140 teams and approximately 2500 female players, 13-19 years old. Due to the ongoing pandemic many players may have quit playing football, and we are currently not aware of that number, but the 400 players will thus represent about 16 % of female players in Stockholm. We have added some of this information in the Methods section:

Football clubs with adolescent female academy players aged 12 to 19 years, participating in Swedish divisions 1-2 for girls in the largest regions, are eligible to participate in the study. The district of Stockholm consists of 140 teams and approximate 2520 female players, 13-19 years old.

5. Will any incentives be provided to clubs and/or players to participate in the study? Given the longitudinal design and respondent burden, incentives could be useful for increasing the quantity and quality of the data obtained.

Response: Before the recruitment of study participants, the research team always have meetings with the parents/guardians to inform about the study aims and implications. In addition to that, and according to Swedish ethical authority guidelines and established Swedish sports medicine research traditions, we do not provide any specific incentives such as money, clothes, equipment, etc. Our shared previous experiences from different projects and settings are that incentives are not needed to recruit and engage participants in research like this in Sweden.

6. Inclusion of the detailed descriptions (and photographs) of the physical tests is very helpful. It would also be helpful to include the questionnaires used to assess the other (e.g., injury history, psychological, social) constructs, as they are not described in detail.

Response: Thank you for appreciating the details and photos. Unfortunately, the questionnaires exist only in Swedish yet, and, even if we agree that this would be ideal, a translation to English (and back-translation) procedure is currently beyond the scope of our study. If the editor so wishes we can provide the forms as online appendix in Swedish.

7. How susceptible are the physical tests to training-related change? Do test values typically change over the course of the season? If so, they should be measured multiple times to obtain greater sensitivity with respect to vulnerability to injury.

Response: We appreciate your point and agree that it would have been optimal. We have a consecutive inclusion of participants, and it is therefore difficult to have both repeated measures in parallel with baseline tests. Importantly, however, in a subgroup of approximately 100 players, all physical tests are repeated after one year to give us information about any changes in test scores and to give a valid risk estimate.

8. Is the single-item perceived stress scale sufficient to account for the “history of stressors” variable that has been so consistently related to the occurrence of sport injury in previous research?

Response: In this multidisciplinary study we have discussed the content and the extent of the content to cover as much as possible. We try to cover as much as possible to without causing survey fatigue. It is not optimal with a single question but according to the literature it is suggested to be enough to capture the perceived stress.

Salminen S, Kouvonen A, Koskinen A, et al. Is a single item stress measure independently associated with subsequent severe injury: a prospective cohort study of 16,385 forest industry employees. *Bmc Public Health* 2014;14(1):543

9. How does the multitude of variables assessed and the prospect of many interactions affect statistical power and sample size calculations?

Response: This is a relevant remark. As we have stated in the Methods section under the subheading “Sample size”, the statistical power for the analyses depends on the exact research question, the number of exposed players, what outcome we will focus on, and if the exposure is continuous or categorised, etc. Importantly, we will not be able to study rare injury locations or rare exposures, and the statistical power will also be limited for interaction analyses.

10. With reference to point 1 above, to what extent will a theoretical framework be used to guide and organize the statistical analysis? Will the statistical analysis take into account the “complex systems approach” of Bittencourt et al. (2016)?

Bittencourt, N. F., Meeuwisse, W. H., Mendonca, L. D., Nettel-Aguirre, A., Ocarino, J. M., & Fonseca, S. T. (2016). Complex systems approach for sports injuries: Moving from risk factor identification to injury pattern recognition – narrative review and new concept. *British Journal of Sports Medicine*, 50, 1309-1314. doi:10.1136/bjsports-2015-095850

Response: Thank you for this important comment. We are aware of that the nature of sport injuries are multifactorial and of complex nature and we aim to study aetiology with consideration taken to interaction on the multiplicative as well as the additive scales. Despite that this cohort study will be among the largest of its kind, we will have limited statistical power to study how factors interact and these analyses will be limited to injuries of any origin, and common exposures. The methods suggested by Bittencourt et al., is one possible way to move from risk factors to risk pattern recognition when appropriate.

Building on the knowledge that the multifactorial complex nature of sports injuries arises not from the linear interaction between isolated and predictive factors, but from the complex interaction among a web of determinants (Bittencourt et al., 2016) a suggested statistical analysis could be CART. This type of analysis also suits research questions related to interactions between independent variables (e.g., from different discipline).

By measuring potential risk factors, confounders and effect modifiers from many areas, including health, lifestyle, socioeconomic factors, football-related factors, psychosocial factors, previous injuries/pain and clinical performance, the bio-psychosocial perspective we will have unusually good conditions to identify independent risk factors, which also is important knowledge in future prevention strategies.

We have added information about this in the section about strengths and limitations

Reviewer: 3

Dr. Dominic Malcolm, University of Loughborough, Comments to the Author:

Overall, this is a well presented study protocol which clearly describes the work being undertaken.

The work requires a more careful proofread as there are a number of sentences which lack clarity.

Response: Thank you for your positive comment. We have re-written the manuscript and changes are in some places marked using “track changes” and it has also been reviewed by an English native speaker.

There is no discussion of how the study is funded.

Response: In the Footnotes, we are describing the received grants so far: Swedish Naprapathic, and the Norwegian Naprapathic Associations, Swedish Research Council for Sport Science, Active life foundation and Sophiahemmet foundation. We will continue to apply for further grants.

There is no discussion of the limitations of this study.

Response: We have in fact mentioned limitations under the heading “Strengths and limitations”. We have followed the authors’ guidelines regarding study protocol provided by BMJ Open. Please see also our response to the editor in the first part of this document.

Finally - and in part these comments relate to what I see as a limitation of the study, but make a more substantive, standalone point - I feel that the description of the study as ‘bio-psychosocial’ is a misrepresentation. For instance, within the list of ‘social factors that influenced the risk of injuries in female athletes’ (page 5) are 3 references to psychological studies of stress (references 20, 29 and 32). The other factors are player/coach education, smoking and playing position.

Subsequently, in the description of the questionnaires used in the KIC there is only one factor (guardians’ education) listed under socioeconomic factors while ‘tobacco as smoking’ is listed under lifestyle and playing position is listed under ‘football-related factors’.

My point is that the ‘social’ in ‘bio-psychosocial’ is not treated with the same level of rigor and depth as the ‘bio’ or the ‘psycho’. You could go into detail about the social determinants of, eg. diet and tobacco smoking (both strongly influenced by ethnicity, gender and social class), but that does not seem to be part of the study. For me the lack of a thorough treatment of the social is a limitation of the study.

Response: Thank you for this important observation. We will use data regarding e.g., diet, tobacco, and parents’ education in our analyses. We are well aware of the problem to give all parts in a biopsychosocial perspective the same attention and value. There are several suggestions for content in a biopsychosocial model and when designing this study, we were influenced by Wiese-Bjornstal (2010). See also our response to comment #1 and #10 from Reviewer 2.

The nature of reviewing protocols is such that the only recommendation can be to re-describe the project. Ensure that what you describe as social really is social, and ensure consistency in describing the social. It should be fundamental to any approach describing itself as bio-psychosocial.

Response: Thank you for this important point. We will base our interpretation of bio-psychosocial perspective on the work of Wiese-Bjornstal. Using a holistic and multidisciplinary approach we are aiming to integrate parts from all disciplines to investigate potential risk factors to understand the complex aetiology. We have considered several biopsychological models e.g., Appaneal & Perna, 2014; Brewer, 2010; Wiese-Bjornstal, 2010 and collected information from them when designing the study.

VERSION 2 – REVIEW

REVIEWER	Malcolm, Dominic Hospital for Special Surgery, Academic Training
REVIEW RETURNED	17-Dec-2021

GENERAL COMMENTS	Thank you for responding positively to my previous comments. I have no further suggestions for changes.
---

REVIEWER	Brewer, Britton Springfield College
REVIEW RETURNED	02-Dec-2021

GENERAL COMMENTS	I appreciate the efforts of the authors in addressing my concerns and those of the Editor and the other reviewers. The quality of the manuscript has been greatly improved as a result of the changes that have been made. The findings of the proposed study are sure to yield valuable insights.
--